# Do powerful CEOs matter for earnings quality? Evidence from Bangladesh

**H. M. Arif, Mohd Zulkhairi Mustapha** ⓘ *, **Azlina Abdul Jalil** ⓘ

Department of Accounting, Faculty of Business and Economics, University of Malaya, Kuala Lumpur, Malaysia

* zulkhairi@um.edu.my

**Data Availability Statement:** Financial data related to measures of earnings quality and control variables of this study have been collected from S&P Capital IQ database, https://www.spglobal.

## Abstract

This study investigates the effects of powerful Chief Executive Officers (CEOs) on earnings quality in a setting where CEOs have strong dominance over other top executives and occasionally attempt to exert their influence over corporate regulatory bodies. Using 10-year longitudinal data for the period from 2010 to 2019 and 1,395 firm-year observations from listed non-financial firms in Bangladesh, we found that CEOs' political power and CEOs with high structural and expert power have a significant detrimental effect on earnings quality. Ownership and prestige power have an insignificant impact on earnings quality. These powerful CEOs use accrual and real activity manipulation techniques together to manage the earnings. This study uses the system-generalized method of moment estimates for estimation purposes, and the results remain robust when alternative earnings quality proxies are used. Taken together, our results suggest that CEOs' political duality (i.e., serving simultaneously as a member of parliament and a CEO) should be restricted and that a CEO's tenure should be limited to a reasonable period. This research adds to the existing body of knowledge by offering empirical support for CEO power dynamics on earnings quality, specifically political and prestige power.

## Introduction

Whether or not powerful CEOs are beneficial or detrimental to earnings quality is an unresolved issue in academic literature and policy formulation. Proponents of powerful CEOs argue that power aligns a CEO's interests with other stakeholders by reducing their myopic practices of misreporting earnings to achieve immediate financial targets [1]. Conversely, other agency theorists posit that excessive power alienates the financial interests of CEOs and general shareholders [2, 3]. It underpins the CEO's position at the apex of concern by impairing the board's supervising effectivity [4]. The practical resonances of this theoretical debate are also evidenced in empirical findings.

A detailed review of the existing literature closely associated with this study reveals that 64% of the prior studies [1, 5–17] (14 out of 22) focused on a developed economy, the US (United States), and that only 32% of prior studies [18–24] (7 out of 22) focused on emerging economies, namely China, Taiwan, Thailand, Vietnam, and Nigeria. One study focused on the

com/marketintelligence/en/. Missing data in the database and other non-financial data have been collected from published and publicly available audited financial statements, annual reports, and prospectuses of listed non-financial firms. These annual reports and prospectuses are available in the websites of the respective firms. The list of non-financial firms is available on the website of Dhaka Stock Exchange, https://www.dsebd.org/by_industrylisting.php. Data related to political connection have been collected from the website of Bangladesh Parliament, available at http://www.parliament.gov.bd/index.php/en/about-parliament/name-and-composition-of-parliament and Bangladesh Election Commission database, http://www.ecs.gov.bd/page/election-results. Data related to elite educational institutions have been collected from the Times Higher Education world university rankings, available at, https://www.timeshighereducation.com/world-university-rankings/2021/world-ranking#!/page/0/length/-1/sort_by/rank/sort_order/asc/cols/stats.

**Funding:** This study received funding from the Special Publication Fund of the Faculty of Business and Economics, Universiti Malaya, in the form of publication fee support. The Universiti of Malaya provided support in the form of salaries for MZM and AAJ. The specific roles of these authors are articulated in the 'author contributions' section.

**Competing interests:** The authors have declared that no competing interests exist.

French context [25]. Thus, there is a shortage of empirical proof regarding the effect of powerful CEOs on earnings quality from emerging nations.

A deeper look reveals that studies that focused on the US economy provided mixed evidence. The results of 71% of previous studies [5–8, 10–14, 17] (10 out of 14) suggest that powerful CEOs are detrimental to earnings quality, whereas 29% of these studies [1, 9, 15, 16] (4 out of 14) suggest that powerful CEOs are associated with improved earnings quality. The inconsistency in empirical findings is much stronger in emerging economies. The empirical results of 43% of studies [18, 19, 23] (3 out of 7) that focused on emerging nations suggest that powerful CEOs are associated with more earnings management. On the other hand, 57% of studies [20–22, 24] (4 out of 7) provide evidence that powerful CEOs are beneficial for reported earnings quality. Thus, the extant literature provides inconsistent findings regarding the effect of powerful CEOs on earnings quality. This inconsistency in findings suggests that there is merit in investigating this relationship. Further, we argue that the existing investigations that attempt to identify the impact of powerful CEOs on earnings quality [5, 9, 12] consider only a portion of structural (CEO pay slice) or ownership power (CEO equity incentives), with considerably more focus on the US (United States) market. These studies ignore other fundamental sources and elements of power; however, the power that is also treated as uneven control of valuable resources in social relationships [13, 14] is much more consequential in emerging economies. The US wealth research firm, Wealth-X, recently revealed that emerging nations are experiencing the world's fastest growth in the millionaire population [26]. Yet, these nations also suffer from substantial income inequality. Income inequality in developed economies is largely a matter of wage disparity. Due to this, several previous studies focused on the CEO pay gap with other top executives as a measure of CEO power, which seems meaningful for the US context. Conversely, in emerging economies, income inequality is an all-pervasive inequality of opportunity. Thus, there is a need to extend the existing literature in the setting of emerging economies.

Recently, a few studies [17, 23, 24] have focused on the influence of powerful CEOs on earnings quality. Our research is distinct from these studies in numerous ways. First, our study focused on CEOs' political and prestige power in addition to structural, ownership, and expert power. While Shiah-Hou [17] focused on structural, ownership, and expert power, Sani, Latif, and Al-Dhamari [24] focused on the ownership, expert, and politically connected board, and Le, Kweh, Ting, and Nourani [23] mainly focused on ownership and expert power. As a result, there is a knowledge gap regarding the effect of CEO political power and prestige power in extant literature. Most prior studies [27–30] that focused on political connections mainly considered politically connected boards but not specifically politically connected CEOs. By focusing on CEOs' direct political connections, this study provides compelling evidence regarding the effect of CEOs' political connections on earnings quality in an emerging economy. Second, our study offers empirical documentation from an emerging economy characterized by a weak regulatory atmosphere. Shiah-Hou [17] provided evidence from a developed economy with a strong regulatory atmosphere (the United States context); although Sani, Latif, and Al-Dhamari [24] and Le, Kweh, Ting, and Nourani [23] focused on the emerging economy context (Nigeria and Vietnam respectively), they offered contradictory results. Sani, Latif, and Al-Dhamari [24] evidenced an enhancement in the quality of earnings with the increase in CEO power, whereas Le, Kweh, Ting, and Nourani [23] documented a decline in the reported earnings quality. This contradiction increased the thirst for further investigation in an emerging economy context. Finally, from a methodological point of view, Shiah-Hou [17] and Le, Kweh, Ting, and Nourani [23] used the fixed-effect estimator, which ignores the dynamic attribute of the relation between powerful CEOs and earnings quality. Our study addressed this issue of dynamic endogeneity by executing a well-established dynamic panel GMM (Generalized

Method of Moments) estimation. This dynamic estimation method also permits to the inclusion of time-invariant effects to check heterogeneity issues. Although Sani, Latif, and Al-Dhamari [24] used GMM estimation, the GMM estimation may be severely biased in small samples if the estimated standard errors are not robust or corrected [31]. Our study addressed this issue by applying Windmeijer [31] corrected standard errors.

We consider Bangladesh a suitable emerging economy for examining the effects of powerful CEOs on the quality of earnings for three reasons. First, Bangladesh has been identified as the world's fastest-growing ultra-high wealthy population (those with a net asset of $30 million or more) over the past five years (2012–2017) [32]. It is predicted to have the third quickest growth in terms of rich individuals (persons with a net wealth of $ 1 million or greater but less than $30 million) in the world in the next five years (2018–2023) [26], and it is a country with high-income inequality that recorded a Gini coefficient at 0.482 in 2016 compared to 0.388 in 1992 [33]. This indicates the existence of severe asymmetric control of resources. Further, increasing corruption and the selective enforcement of laws [28] make it easy for powerful individuals to commit opportunism at the cost of general shareholders. For instance, the position of Bangladesh was 146 in the 2020 corruption perception index, two steps lower than its 2017 position [34].

Second, manipulated corporate earnings reports have become an issue of extreme concern in Bangladesh. For instance, in an interview with a daily newspaper, the chairperson of the Financial Reporting Council (FRC) mentioned that the government recently formed the FRC after the identification of serious manipulation of many firms' financial disclosures [35]. Due to these material misstatements, thousands of general investors and financial firms that provide credit facilities to manipulated firms lose their investments every year. One example is when the earnings manipulation by GMG Airlines was reported, a few thousand shareholders lost BDT (Bangladesh taka) 3,000 million, and a state-owned commercial bank lost BDT 2,470 million [36]. The country is also losing foreign investment in the stock market due to a lack of trust or confidence in listed firms' financial statements [35].

Third, we contend that powerful CEOs in Bangladesh perform a major role in the systematic change in earnings quality. Our data shows that more than half the CEOs of listed non-financial firms are close relatives to board chairs (e.g., son, daughter, brother, sister, husband, or wife). Eventually, these CEOs have extreme control over corporate internal governance. On top of that, the exercise of political power in the corporate environment by CEOs has gained momentum. Since Bangladesh's independence in 1971, 11 parliamentary elections have been held. In the first parliament (1973–1979), businessmen comprised 18% of the parliament's members, but in the ninth (2009–2014), tenth (2014–2018), and eleventh parliaments, businessmen represented 57%, 59%, and 61% of the members, respectively [37]. Muttakin, Monem, Khan, and Subramaniam [28] argued that political allies to the party in power are the precondition necessary to protect and advance the economic interest of business leaders in Bangladesh. For example, only one pharmaceutical company is the exclusive distributor of Covishield (Covid-19 vaccine) in Bangladesh. From January to March 2021, the firm distributed 5 million doses of vaccines (out of 30 million doses) and made a net profit of BDT 383.7 million, which is almost 62.5% higher than the same period of the previous year [38]. Ultimately, the firm will gain a net profit of BDT 2,310 million from the distribution of 30 million doses. Interestingly, the CEO and the vice-chairman of the company are both parliament members from the ruling party, and the vice-chairman also holds the rank of a cabinet minister for his advisory role to the prime minister [39]. It has been alleged that the firm pressured the government not to adopt alternative sources of vaccines [39]. Hence, this study focuses on CEOs' political power along with the forms included in Finkelstein's [40] taxonomy of power.

This research initiates several contributions. First, our investigation strengthens Finkelstein's [40] typology of power by adding the CEO's political power as a source of corporate leaders' strength and providing empirical attestation regarding the impact of CEOs' political and prestige power on earnings quality. Second, our study sheds light on the debate in agency theory regarding the implication of powerful CEOs on earnings quality in emerging economies by clarifying the mixed findings and providing empirical evidence from Bangladesh. Third, our study addresses the issue of dynamic endogeneity in addition to heteroscedasticity and serial correlation which was ignored in most of the earlier studies. We used the system-GMM estimation which provides consistent results and alleviates endogeneity concerns in existing findings. Scholars suggest that neglecting the dynamic endogeneity may create profound consequences on inference accuracy [41].

Finally, we use another metric, that of elite educational institutions, which is an element of prestige power that differs from Finkelstein [40] and Lisic, Neal, Zhang, and Zhang [42]. Our measurement addresses the shortcomings of previous attempts to include the metric, as those earlier studies identified only universities from the United States as elite and ignored other top-ranked universities from the rest of the world. They also ignored that university prestige or rank is not fixed and varies over the years. Our study considers the top 300 universities every year that are ranked by the Times Higher Education World University Overall Rankings as elite educational institutions.

## Corporate governance regulations in Bangladesh

The first edition of Corporate Governance Guidelines (CGG) was released by the Securities Commission of Bangladesh in 2006. However, compliance with the guidelines was not mandatory for listed firms [43]. After a massive stock market crash in 2010, the commission amended the CGG in 2012; it contained 95 conditions under seven headings and one annexure [44]. This time compliance with the guidelines was mandatory. In 2018, the commission replaced the CGG with the Corporate Governance Code (CGC), which contains 166 conditions under nine headings and three annexures [45].

Although the CGC has expanded almost 75% in terms of the conditions it enforces, it provides enough space for CEOs to exercise extreme power. For instance, both the latest code and the previous CGG restrict CEO duality; but most nonfinancial firms follow this condition in the form only, but not in substance. Our data shows that 52% of our sample firms have CEOs and board chairs from the same family, and 77% of sample firms have the CEO as the only top management representative who sits on the board. Furthermore, the code is silent regarding the maximum number of committees of a board upon which a CEO can be a member, the maximum tenure of office, and CEOs' political affiliations. Our data also reveals that each CEO possesses on average 6.89% of their firm's total equity, and their family members possess on average an additional 16.98%. Moreover, CEOs adopt the position of CEO on average for 11 years, and each CEO secures directorship on 8 other corporate boards. Corporate boards include both public and private limited firms.

## Review of literature

Extant literature does not offer a single universally accepted definition of earnings quality. For example, Neel [46] mentioned that high-quality earnings are those that are presented faithfully. In other words, quality earnings should be presented in a neutral, complete, and free-from-error condition [46]. According to Dechow, Weili, and Catherine [47], the quality of earnings depends on its' decision usefulness. In other words, the higher the relevance of the earnings figure for any specific decision, the higher the quality of that earnings. Accordingly, Dechow,

Weili, and Catherine [47] defined quality earnings as "higher quality earnings provide more information about the features of a firm's financial performance that are relevant to a specific decision made by a specific decision-maker". According to Dichev, Graham, Harvey, and Rajgopal [48] high-quality earnings are backed by actual cashflows, and accurately reflect the results of the operation, and the economic reality of the firm. They also suggested that consistent reporting choices over time and the absence of one-time items improve the level of earnings quality. This study assumes that high-quality earnings are the results of low-level earnings management practices [47]. Earnings management results from the managers' use of accounting choices or making operational decisions to hide the actual operating performance [49]. In other words, high-quality earnings should contain a low level of abnormal accruals that result from the discretionary use of accounting rules [47] and should reflect a low level of real activities management. Based on this concept of earnings quality, following Hope, Thomas, and Vyas [50], this study uses the accrual models to measure earnings quality. Dechow, Weili, and Catherine [47] suggest that the application of accrual models as proxies for earnings quality has become a recognized practice in the accounting literature.

Power is treated as the capability of a person to enforce his or her desire and influence the behavior of others [40]. Finkelstein [40] classified organizational power into four clusters: structural, expert, ownership, and prestige power. Additionally, our study considers the political ties of CEOs as a significant source of power in the context of the emerging economy. This study views the problem of poor-quality financial reporting via the lens of agency theory (Type II agency problem). In Bangladesh, where a large portion of the non-financial listed firms are family-dominated [51] and more than half the CEOs are family members, firms experience greater Type II agency conflicts (disputes of interests within controlling inside shareholders and minority investors) than Type I. These CEOs act as representatives of other family members on the board, take opportunistic decisions, and extract personal gains through channels such as excessive remuneration, special dividends, advantageous related-party transactions, and sometimes even by gaming the system established to monitor the interests of dispersed investors [52]. Eventually, the Type II agency conflicts reduce the earnings quality [53].

## CEOs structural power and earnings quality

A part of a CEO's power stems from the formal position the CEO holds within a firm. Scholars argue that a manager with a legitimate right to exert influence is influential [40, 54] as it allows for a CEO's substantial discretion over the allocation of resources [55]. Excessive concentration of structural power, which is founded on organizational structure and hierarchy of power for CEOs, facilitates self-enhancement attempts among CEOs with self-serving motives [56]. This in turn reduces reported earnings quality. Scholars argue that a CEO's position itself is a basis of structural power [57]. Furthermore, the addition of other elements of dominance, such as being the chair of the board, being the only top management representative on the board, and being a member of different board committees, adds stronger layers to the structural power of a CEO. This high level of structural power opens the door for the CEO to create undue pressure on other top executives for earnings management to inflate earnings [58].

However, whether the excessive structural power of the CEO is beneficial or harmful to firms is an unresolved question. Empirical studies show both positive and negative impacts on a firm's financial reporting quality. For example, Hu and Gan [19] noted that CEOs' greater structural power enhances the firms' internal control quality. Tee [59], in research focusing on the Malaysian context, mentioned that powerful CEOs boost the quality of accrual income. Also, Meo, Lara, and Surroca [1] and Seifzadeh, Rajaeei, and Allahbakhsh [60] identified an inverse tie between entrenched managers and income distortion. Contrarily, Shiah-Hou [17]

and Baker, Lopez, Reitenga, and Ruch [11] mentioned that CEOs' greater structural power raises accrual earnings management. Koo and Kim [5] also identified an increase in firms' opacity parallel to an increase in CEOs' structural power. Nuanpradit [18] and Bouaziz, Salhi, and Jarboui [61] documented a supportive nexus between CEO duality and accrual earnings manipulation. In the context of Bangladesh, Muttakin, Khan, and Mihret [62] documented a statistically significant and positive impact of CEO duality (an element of structural power) on accrual earnings management. Mande and Son [7] mentioned that high structural power increases the likelihood of beating or meeting market experts' forecasts. So, there is a lack of abstract knowledge regarding the consequence of the high structural power of CEOs on earnings quality in the existing literature.

In the context of Bangladesh, CEOs with high structural power enjoy unrestricted decision-making power and have complete access to all inside information [63]. Muttakin, Khan, and Mihret [64] argued that CEOs in Bangladesh are mainly family CEOs, and they are less concerned about legitimacy and accountability. Other top executives enjoy very little formal power within the organization compared to CEOs. Normally, they do not have membership in the board of directors. Joseph, Ocasio, and McDonnell [65] argued that the position of CEOs as insiders makes them information brokers and allows them to remove potential internal challengers. Thus, this study expects that excessive structural power will pave the way for CEOs to dominate boards and create pressure on other top executives to harvest short-run stock market gains through manipulating earnings. In Bangladesh, where businesses face a significant lack of regulatory predictability, with powerful business leaders involved in serious corruption [34], there is a favorable environment for abusing excessive power. Therefore, we conjecture the following hypothesis:

Hypothesis 1: CEOs with greater structural power will be negatively associated with earnings quality

## Ownership power of CEOs and earnings quality

Scholars argue that managerial shareholdings reduce board influence and a manager with a significant shareholding will be more dominant than a manager without such ownership [40, 66]. Agency theorists argue that ownership assists to conform the interests of principals and agents [2]. As per this argument, CEOs' stock ownership should reduce earnings manipulation [67]. However, the excessive share ownership of CEOs may also intensify the Type II agency conflicts that ultimately decrease the earnings quality [53, 68, 69]. Extent literature that focused on CEOs' shareholdings and earnings quality has found inconclusive results. Several studies mentioned that CEO ownership power enhances earnings quality [9, 17, 24, 67, 70]. However, several studies showed evidence that CEO ownership power reduces internal control quality [19] and intensifies earnings fabrication [12, 71]. Further, several studies found an insignificant effect. For example, using samples from listed Vietnamese firms, Le, Kweh, Ting, and Nourani [23] identified an insignificant association between CEO ownership power and earnings management. Also, Hribar and Nichols [72] found an insignificant effect of managerial equity incentives on earnings management after controlling the standard deviation of sales and cashflows. Rashid [73], in the context of Bangladesh, identified a statistically insignificant association between managerial ownership and agency costs. Thus, whether CEOs with high ownership power are associated with improved earnings quality or decreased quality of earnings is an unsettled question that demands further investigation. Considering the ownership structure in Bangladesh, where more than 70% of firms are controlled by family members [51, 74] (a signal of severe Type II agency problems), this research anticipates a negative association between ownership power and earnings quality. So, this study makes the following assertion:

Hypothesis 2: CEOs with greater ownership power will be negatively associated with earnings quality

## CEO expert power and earnings quality

Finkelstein [40] defined expertise as the capability to deal with environmental contingencies. Following Finkelstein [40], this study focuses on CEOs' functional expertise since functional expertise also opens the door to enjoying decision-making autonomy and practicing managerial autocracy [75, 76]. This study considers a CEO's tenure and professional degree as indicators of functional expertise. Scholars have argued that CEOs' power and the probability of managerial opportunism increase over time [1]. Earlier studies mentioned that managers with longer tenure and more knowledge obtain more control over a company's operational strategies, which induces them to be involved in more earnings manipulation activities [20, 40]. Long tenure also makes it easier for CEOs to establish good relationships with other top executives. As their tenure increases, CEOs find it simpler to put initiatives into action that align with their targets since other senior executives are also motivated by personal financial gain [77, 78]. Furthermore, long tenure helps CEOs establish control over internal monitoring mechanisms [77, 79]. In Bangladesh, there are instances where the CEO himself becomes part of the monitoring mechanism by being a member of the audit committee. Thus, in these cases, the monitoring section of a firm plays a ceremonial role. This paves the way to engaging in self-serving behaviors detrimental to minority shareholders.

Besides tenure, CEOs with professional accounting or finance qualifications are very much familiar with the firms' financial reporting process and favorable accounting treatments [80]. Therefore, they can strategically employ different earnings management devices to convey a preferred financial report [73, 81]. Prior studies considering the Bangladesh context documented that firms' accounts departments are acquiescent to the wishes and decisions of powerful chief executives, who instruct based on their interests and desires [63, 82]. Uddin [82], referring to the Bangladesh context, documented that CEOs play a key role in preparing the sales budget, and production managers remain ready to execute their instructions. The decision of the CEO is final regarding any production issue. This situation also creates ample opportunity for CEOs to get involved in real activities management through abnormal production costs.

The extant literature provides mixed findings regarding CEOs' expert power and earnings quality. Several studies mentioned that expert CEOs are fundamental for better earnings quality [16, 24, 83, 84]. In their most recent study, Altarawneh, Shafie, Ishak, and Ghaleb [85], found that long-serving CEOs are linked to higher-quality financial reporting. However, other studies suggest that expert CEOs reduce firms' reported earnings quality [17, 19, 22, 23, 75, 86]. Further, Ferramosca and Allegrini [81] documented an insignificant consequence of expert CEOs on earnings quality. Based on the contention that managerial expertise augments managerial power [22, 40] and that power eventually stimulates the desire to manage earnings [22], this study posits that CEOs with greater expertise will have an adverse consequence on earnings quality. Accordingly, our hypothesis is as follows:

Hypothesis 3: CEOs with greater expert power will be negatively associated with earnings quality

## Prestigious CEOs and earnings quality

Prestige is treated as the respect one possesses in the eyes of others [55]. Finkelstein [40] argued that managerial prestige is based on a person's reputation in institutional environments, social organizations, and among stakeholders. Also, managers' elite educational

backgrounds are an important source of prestige [40, 42]. Attending elite educational institutions paves the way to familiarizing themselves with other business elites and creating useful networks of expertise [87]. Scholars suggest that prestige intensifies managerial power by absorbing uncertainties from the institutional environment [55, 40]. Further, prestige power induces CEOs to take risky initiatives since their social ties help to get new ideas, protect jobs, and improve the chances of re-employment after the CEO is fired for failed risky strategies [88–90]. We expect that as CEOs with high-prestige power sit on numerous corporate boards and maintain strong social networks, it is easy for them to manipulate revenues and expenses if required. Fang, Francis, Hasan, and, Wu [91] argued that the reciprocal relationship of CEOs with other corporate boards helps them to initiate earnings management by scheduling sales and expenses, employing relationship sales, or introducing asset exchange transactions.

Empirical investigations closely associated with our study provide very limited and inconclusive findings about the implication of CEOs with high prestige on earnings quality. A few studies showed evidence that CEOs with greater connections maintain a high quality of accrual earnings [85, 92], and these CEOs are associated with minimal fraudulent incidents [93]. However, some other studies demonstrate a positive association between well-connected CEOs and earnings management [88, 91, 94]. These minimal and inconclusive findings demand more inquiries as to the effect of CEOs' prestige power on earnings quality. This study posits the following hypothesis:

Hypothesis 4: CEOs with greater prestige power will be negatively associated with earnings quality

## CEO political power and earnings quality

Scholars argue that CEOs' political affiliation makes it easy to access financial resources and get favorable regulatory treatment [95]. Hence, the political connection is associated with improved firms' performance. However, some other studies suggest that CEOs' political connections may weaken the effectiveness of the managerial supervision system and curtail the chances of politically linked CEOs being sacked [96]. Also, political connections work as security against punishments for low-quality accounting information [27]. In some cases, these CEOs are not penalized because it may negatively impact the political party's image. Eventually, these CEOs become entrenched on the board and other top executives and have a detrimental effect on the quality of information they disclose. Extent literature provides limited and inconclusive findings regarding this issue. Previous investigations mainly considered politically connected boards and not politically connected CEOs. Some studies show that firms with political associations are associated with better earnings informativeness [97], better firm accomplishment [28], and improved value of entities [98]. In contrast, some other studies identified poor quality earnings in politically linked firms [27, 29, 95]. Even, Liu, Li, Zeng, and An [99] documented an inverse U-shaped association within the politically connected firms and transparency in accounting. These indeterminate findings urge further investigation into the politically linked CEOs' effect on earnings quality.

In the context of Bangladesh, politically connected CEOs do not have to care about rules and regulations as rules and regulations are perceived for managers with low power. Ahmed and Uddin [63] mentioned that politically connected business leaders in Bangladesh are usually able to bypass or avoid reasonable initiatives taken by state regulatory authorities. The extent of following corporate governance codes or guidelines is up to their comfort level [63]. If they feel any prospective negative outcome for themselves by following a particular instruction from a regulatory body, they simply ignore it or even mention that they executed the instruction without following it [63]. In the context of Bangladesh, Sobhan [100] found that

annual reports significantly overstated the extent to which corporate governance guidelines were followed. We anticipate that in an emerging economy like Bangladesh, where the competition to access resources is very high and there are instances of selective enforcement of laws, CEOs' political connections stimulate them to manipulate earnings to show better firm achievements. Hence, we postulate the following hypothesis:

Hypothesis 5: CEOs with political power will be negatively associated with earnings quality

## Research approach

### Sampling and data

This research utilized a ten-year unbalanced panel data set from 165 firms enlisted on the Dhaka stock exchange over the years 2010 to 2019. As of June 30, 2019, a sum of 317 firms is enlisted in the Dhaka Stock Exchange [101]. First, we excluded firms in the bank, insurance, and other financial sectors because these firms are subject to special regulatory treatments and they have specific accounting practices [102, 103]. Next, we excluded the firms in the industry with less than 8 business entities to confirm enough data to measure the earnings quality proxies [104]. Finally, this study excluded firms with insufficient financial data. The sample of this study comes from 8 industries. These are textile, engineering, food and allied, IT and telecom, fuel and power, cement and ceramics, pharmaceuticals, and miscellaneous. An ultimate sample size of 1,395 observations from 165 firms is shown in Table 1.

We used the audited financial statements to get the financial data of our sample firms, which had been published in their annual reports and from the database S&P Global Market Intelligence. This study found non-financial data related to different dimensions of CEO power mainly from publicly available annual reports, prospectuses, and websites of the sample firms. To identify the CEOs' political connections, this study obtained data from the websites of the Bangladesh Election Commission and Bangladesh Parliament in addition to annual reports.

### Earnings quality measures

To measure earnings quality, consistent with Hope, Thomas, and Vyas [50] we applied two of the most popular proxies of accrual income. Our initial proxy of accrual quality was derived using the revised version of the Dechow and Dichev [105] model. This study, following prior studies, estimated the following equation cross-sectionally for each year and every industry, with a minimum of 8 firm-year observations to confirm enough data to capture the estimated parameters [104, 106, 107]. Each of the constituents in this equation is scaled by the lag total assets.

$$TAC_{it} = \gamma_0 + \gamma_1 CFO_{i,t-1} + \gamma_2 CFO_{it} + \gamma_3 CFO_{i,t+1} + \gamma_4 \Delta Rev_{i,t} + \gamma_5 PPE_{it} + \gamma_6 DCFO_{it} + \gamma_7 CFO_{it} \\ \times DCFO_{it} + \epsilon_{i,t} \tag{1}$$

**Table 1. Sample selection.**

| | |
|---|---|
| Total Firms Listed on Dhaka Stock Exchange on June 30, 2019 | 317 |
| Less: Banks, Insurance, and Financial Institutions | 102 |
| Less: firms in the industry with less than 8 firms | 21 |
| Less: firms with insufficient financial data | 29 |
| Number of firms treated as the final sample | 165 |
| Number of observations | 1,395 |

Where:

| | | |
|---|---|---|
| $TAC_{it}$ | = | total current accruals calculated as ($\Delta CA_{it}-\Delta CL_{it}-\Delta Cash_{it}+\Delta Debt_{it}$) |
| $\Delta CA_{it}$ | = | yearly variation in total current assets |
| $\Delta CL_{it}$ | = | yearly variation in total current liabilities |
| $\Delta Cash_{it}$ | = | yearly variation in total cash |
| $\Delta Debt_{it}$ | = | yearly variation in total current debt |
| $CFO_{it}$ | = | represents cash flows from operating activities |
| $\Delta Rev_{it}$ | = | yearly variation in sales revenues |
| $PPE_{it}$ | = | represents fixed assets (on a gross basis) |
| $DCFO$ | = | 1 for negative cash flows from operation, and 0 otherwise |

Discretionary total current accruals (DisTAC) are the residuals from Eq 1. Following Hope, Thomas, and Vyas [50], this study first generates the absolute values of these residuals and then multiplies these absolute values by -1. Thus, the bigger value of DisTAC, the better the quality of accrual income.

Our second proxy was derived applying the performance-reconciled model of discretionary accruals recommended by Kothari, Leone, and Wasley [108]. This study, following prior studies, estimated the following equation cross-sectionally for each year and every industry, with a minimum of 8 firm-year observations to confirm enough data to capture the estimated parameters [104, 106, 107]. Each of the constituents in this equation is scaled by the lag total assets.

$$TA_{it} = \gamma_0 + \gamma_1(1/Assets_{i,t-1}) + \gamma_2(\Delta REV_{i,t}) + \gamma_3 PPE_{i,t} + \gamma_4 RoA_{i,t} + \epsilon_{i,t} \qquad (2)$$

Where:

| | | |
|---|---|---|
| $TA$ | = | total accruals measured as ($\Delta CA_{it}-\Delta CL_{it}-\Delta Cash_{it}+\Delta Debt_{it}-DEPN_{i,t}$) |
| $DEPN_{it}$ | = | represents annual reduction in tangible and intangible assets |
| $PPE$ | = | the worth of fixed assets (on a net basis) of a firm |
| $RoA$ | = | the proportion of net operating income to average assets |

Discretionary total accruals (DisTA) are the residuals from Eq 2. As per Hope, Thomas, and Vyas [50], we first generated the absolute values of these residuals and then multiplied these absolute values by -1. Therefore, the larger value of DisTA, the better the quality of accrual income.

## Measures of CEO power dimensions

**Structural power.** To measure CEO structural power, this study considered three elements—CEO's board committees, CEO duality, and CEO only insider. This study began by keeping track of how many board committees the CEO participates in. A CEO who serves on several board committees may have more power and the potential to influence the board's important decisions due to the concentration of decision-making in one person [109]. Eventually, the variable CEO's board committees get the score of 1 if a CEO's presence on corporate board committees is above or the same as the sample median, or 0 if otherwise [109]. The use of the sample median as a benchmark to decide whether any element generates power for a CEO has been documented in prior studies [42, 64, 110]. Next, CEO duality gets a score of 1 if

the board president and CEO are the same person or 0 if otherwise [42]. Finally, CEO-only insider gets a score of 1 if CEO is the only person who represents top management in board meetings, or 0 if otherwise [111]. This study then generates CEO structural power (Strct_pwr) by adding CEO duality, CEO-only insider, and CEO's board committees.

**Ownership power.**   Finkelstein [40] suggested that top executives' ownership power stems from both their stock ownership and their family members' stock ownership. Therefore, to measure CEO's ownership power, this study considers two elements: the percentage of the CEO's share ownership and the percentage of the CEO's family members share ownership, excluding the CEO's portion. A father, mother, brothers, sisters, spouses, and children are treated as family members. This study assumes that if either a CEO or his or her family members hold a substantial portion of stock ownership, the CEO enjoys the full advantages of ownership power. Thus, CEO ownership power (Own_pwr) is a dummy variable that gets a score of 1 if the percentage of the CEO's share ownership or the percentage of the CEO's family ownership is above or the same as the sample median, or 0 if otherwise.

**Expert power.**   To measure CEO expert power, this study considered two elements: CEO tenure and CEO professional degree. This study uses the length of the CEO's tenure as a source of expert power because the ability to handle environmental uncertainties and the competence to influence a firm's key strategic decisions are likely to increase with greater tenure [42, 109]. Also, a CEO's accounting and financial expertise is revealed by a degree such as chartered accountant, cost and management accountant, or certified financial analyst [22]. This study assumes that if a CEO holds the post for a long time or achieves a professional accounting or finance degree, he or she can enjoy the privileges of expert power. Therefore, CEO expert power (Expt_pwr), which is a dummy variable, gets a score of 1 if the year of experience as a CEO is greater than or equal to the sample median, or if the CEO has a professional degree (CA or CMA or CFA), or 0 if otherwise.

**Prestige power.**   Following earlier studies [40, 42, 110], to measure CEOs' prestige power, this study considers three elements of prestige power. These are CEOs' other corporate board directorships, nonprofit board directorships, and elite education (educational background). Corporate board directorship gets a score of 1 if the number of corporate boards a CEO serves on is above or the same as the sample median, and 0 if otherwise. Non-profit board directorship gets a score of 1 if the number of non-profit boards on which a CEO serves is above or the same as the sample median, or 0 if otherwise. To identify elite educational institutions, this study followed the university rankings by the Times Higher Education. This study treats the top 300 universities each year as elite educational institutions. This is because the government of Bangladesh accepts the top 300 universities by Times Higher Education for offering the most prestigious Prime Minister Fellowship [112]. Elite education gets a score of 6 (most prestigious) if a CEO graduated from a university that was ranked within 1–50; 5 if the university was ranked within 51–100; 4 if the university was ranked within 101–150; 3 if the university was ranked within 151–200; 2 if the university was ranked within 201–250; 1 (least prestigious) if the university was ranked within 251–300; or 0 if otherwise. This study then generated CEO prestige power (Prtge_pwr) by adding corporate board membership, non-profit board membership, and elite education.

**Political power.**   CEO political power (Polit_pwr) gets a score of 1 if the CEO, at some point between 2010 and 2019, was a minister, a member of parliament, a government official, held a top post in the ruling party, or was a close relative to a politician, or 0 if otherwise [27, 28].

**Control variables.**   Several company-specific characteristics that have been demonstrated to influence earnings quality were controlled to have a precise understanding of the impact of our independent variables. These include firm size (Firm_siz), leverage (Levge), growth in

revenues (Growth_Rev) [50]; return on assets (RoA) [113]; Big 4 auditor (Big 4) [20]; negative earnings (NegEarn), firm age (AgE) [51]; board size (Bd_siz), institutional ownership (Inst_Own), board independence (Bd_Ind) [83]. To manage the effect of industry and year, year, and industry dummies are added to all estimates [50, 113].

**Estimation method and model.** This research applies the two-step system GMM introduced by Blundell and Bond [114] to assess the link between CEO power and earnings quality because the panel data for his study is dynamic. Prior studies mentioned that the dynamic panel GMM estimator yields consistent outcomes by controlling dynamic endogeneity, heterogeneity, and simultaneity [41, 115]. The dynamic endogeneity, also known as dynamic panel bias, is controlled in GMM by using lags of the dependent variables as regressors [115–117]. So, the lag-dependent variables work as an instrument to control the endogenous relationship. In GMM, the endogeneity of regressors is removed by internally transforming the variables. The first difference and the second-order transformation are two different types of transformation. The previous value of a variable is subtracted from its present value in the first difference transformation [115, 116]. However, the problem with this transformation method is the chance of data loss if the panels are unbalanced and there are missing values [115, 117, 118]. Arellano and Bover [119] developed the forward orthogonal deviation method to address the issue, in which the mean of future available values of a variable is subtracted from the current value. Arellano and Bover [119] proved that this transformation removes the unobserved fixed effects and at the same time preserves the orthogonality among the transformed errors.

This study uses the xtabond2 codes written for statistical software STATA by Roodman [116] to get system GMM estimators for our unbalanced panels. In all the estimations, the Windmeijer [31] correction to the standard errors is used to get more precise estimates. Scholars have suggested that without this correction, reported standard errors will be severely biased [116, 31]. This study utilizes a one-period lag of earnings quality proxies to complete the dynamic process, as lags beyond one period show an insignificant coefficient. This study applies only internally generated instruments that are lag of instrumented variables.

Consistent with prior studies, this study applies two post-estimations to test the appropriateness of the econometric model used: the Arellano-Bond test and the Hansen $J$ test [41, 115]. The use of exogenous instruments is one of the key conditions for the validity of GMM outcomes. The Hansen $J$ test is employed to determine whether the instruments are exogenous or not. Here the instruments' exogeneity is assumed in the null hypothesis [41]. An insignificant test result suggests that instruments are correctly specified. Additionally, there must not be any serial correlation in the idiosyncratic disturbance term for the validity of GMM results [41, 115, 120]. If this assumption is still valid, according to Wintoki, Linck, and Netter [41], the residuals in the first difference AR (1) should be correlated. However, the second differences' residuals AR (2) need to be uncorrelated. The test findings of the Arellano-Bond test and Hansen $J$ statistics are presented after each estimation for this study. To neutralize the impact of outliers in our data set, all continuous variables have been winsorized at the 1% and 99% levels [121–123].

Model 1

$$
\begin{aligned}
EQ_{it} = {} & \gamma EQ_{it-1} + \alpha_1 Strct\_pwr_{it} + \alpha_2 Own\_pwr_{it} + \alpha_3 Expt\_pwr_{it} + \alpha_4 Prtge\_pwr_{it} + \alpha_5 Polit\_pwr_{it} \\
& + \alpha_6 Firm\_siz_{it} + \alpha_7 Levge_{it} + \alpha_8 RoA_{it} + \alpha_9 AgE_{it} + \alpha_{10} NegEarn_{it} + \alpha_{11} Big4_{it} \\
& + \alpha_{12} Growth\_Rev_{it} + \alpha_{13} Bd\_siz_{it} + \alpha_{14} Bd\_Ind_{it} + \alpha_{15} Inst\_Own_{it} + \eta_t + \varphi_i + \upsilon_{it}
\end{aligned}
\tag{3}
$$

Where:

| | | |
|---|---|---|
| $EQ_{it}$ | = | denotes earnings quality proxies (DisTA, DisTAC, DisREV, and Ab_P_C) |
| $EQ_{it-1}$ | = | means the prior year value of the respective earnings quality proxy |
| Strct_pwr | = | structural power (CEO duality + only insider + presence in board committees) |
| Own_pwr | = | ownership power (1 if the percentage of a CEO's stock ownership or the percentage of a CEO's family members' ownership is greater than or equal to the sample median, or 0 otherwise). |
| Expt_pwr | = | expert power (1 if the year of experience as a CEO is greater than or equal to the sample median or if a CEO possesses a professional degree (CA or CMA or CFA), or 0 otherwise). |
| Prtge_pwr | = | prestige power (CEO's other corporate board directorship + non-profit board directorship + elite education) |
| Polit_pwr | = | political power (1 if the CEO is a minister or member of parliament or government official or close relative to a politician or top post holder in the ruling party, or 0 otherwise). |
| Firm_siz | = | total assets of a firm (in natural log form) |
| Levge | = | the proportion of total debt to total assets |
| RoA | = | proportion of net operating income to average assets |
| AgE | = | total years since the company is founded (in natural log form) |
| NegEarn | = | 1 if the firm records a net loss in a year, or 0 if otherwise |
| Big 4 | = | 1 if a firm's financial reports are attested by a Big 4 audit firm, or 0 if otherwise |
| Growth_Rev | = | percentage change in revenues from the immediate past year |
| Bd_siz | = | board size (number of directors) |
| Bd_Ind | = | board independence (proportion of independent directors) |
| Inst_Own | = | institutional ownership (proportion of a firm's share possessed by institutional investors) |
| $\eta_t$ | = | time-specific effect |
| $\varphi_i$ | = | industry-specific effect |
| $v_{it}$ | = | the error term |

## Results

### Univariate and bivariate analysis

Table 2 displays the descriptive stats concerning different dimensions of the CEO's power, financial variables, and earnings quality proxies included in our empirical analyses. Table 2 shows that structural power has a minimal score of 0, maximum 3, with a median figure of 2. Ownership, expert, and political power are treated as binary variables. Prestige power varies between 1 and 8, with a median score of 2. Our data shows that out of 1,395 observations, 77% of the observations have CEOs with high ownership power, while this rate is 55% for experts

**Table 2. Descriptive statistics.**

| Variables | Obs. | Mean | SD. | Min | 50% | 75% | Max |
|---|---|---|---|---|---|---|---|
| Strct_pwr | 1395 | 1.73 | .532 | 0 | 2 | 2 | 3 |
| Own_pwr | 1395 | .771 | .420 | 0 | 1 | 1 | 1 |
| Expt_pwr | 1395 | .546 | .498 | 0 | 1 | 1 | 1 |
| Prtge_Pwr | 1395 | 2.248 | .489 | 1 | 2 | 2 | 8 |
| Polit_pwr | 1395 | .253 | .435 | 0 | 0 | 1 | 1 |
| NegEarn | 1395 | .091 | .288 | 0 | 0 | 0 | 1 |
| Big 4 | 1395 | .209 | .407 | 0 | 0 | 0 | 1 |
| Levge | 1395 | .512 | .474 | .003 | .459 | .658 | 7.516 |
| AgE (log of firm's age) | 1395 | 3.024 | .610 | 1.386 | 3.045 | 3.497 | 4.234 |
| Growth_Rev | 1395 | 11.695 | 25.380 | -55.031 | 8.291 | 22.017 | 73.605 |

(*Continued*)

**Table 2.** (Continued)

| Variables | Obs. | Mean | SD. | Min | 50% | 75% | Max |
|---|---|---|---|---|---|---|---|
| Firm_siz (log of total assets) | 1395 | 21.673 | 1.519 | 17.197 | 21.651 | 22.583 | 26.022 |
| Bd_siz | 1395 | 7.285 | 2.163 | 4 | 7 | 10 | 14 |
| Bd_Ind | 1395 | 20.1 | 11.40 | 0 | 20 | 25 | 60 |
| Inst_Own | 1395 | 15.98 | 10.61 | 0 | 15.97 | 23.15 | 38.11 |
| RoA | 1317 | .055 | .091 | -1.073 | .045 | .085 | .626 |
| DisTA | 1317 | -.157 | .253 | -4.412 | -.098 | -.045 | 0 |
| DisREV | 1317 | -.040 | .081 | -1.995 | -.023 | -.012 | 0 |
| DisTAC | 1252 | -.098 | .193 | -4.687 | -.061 | -.025 | 0 |
| Ab_P_C | 1217 | .803 | .720 | -.031 | 0.603 | 0.952 | 4.200 |

**2.1. Descriptive statistics of the elements of structural power**

| Variables | Obs | Mean | SD. | Min. | 50% | 75% | Max |
|---|---|---|---|---|---|---|---|
| CEO Duality | 1395 | 0.059 | 0.235 | 0 | 0 | 0 | 1 |
| CEO Only Insider | 1395 | 0.680 | 0.467 | 0 | 1 | 1 | 1 |
| CEO's Board Committees | 1395 | 1.207 | .536 | 0 | 1 | 1 | 6 |

**2.2. Descriptive statistics of the elements of ownership power**

| Variables | Obs | Mean | SD. | Min. | 50% | 75% | Max |
|---|---|---|---|---|---|---|---|
| CEO's stock ownership | 1395 | 6.891 | 7.136 | 0 | 4.6 | 10.76 | 21.85 |
| CEO's family ownership | 1395 | 16.979 | 17.824 | 0 | 11.58 | 31.59 | 49.78 |

**2.3. Descriptive statistics of the elements of expert power**

| Variables | Obs | Mean | SD. | Min | 25% | 50% | 75% | Max |
|---|---|---|---|---|---|---|---|---|
| CEO Tenure | 1395 | 11.17 | 9.08 | 1 | 4 | 9 | 16 | 37 |
| CEO Professional Degree | 1395 | 0.022 | 0.145 | 0 | 0 | 0 | 0 | 1 |

**2.4. Descriptive statistics of the elements of prestige power**

| Variables | Obs | Mean | SD. | Min | 25% | 50% | 75% | Max |
|---|---|---|---|---|---|---|---|---|
| Corporate board directorship | 1395 | 8.249 | 10.01 | 0 | 1 | 5 | 11 | 31 |
| Nonprofit board directorship | 1395 | .705 | 1.232 | 0 | 0 | 0 | 1 | 4 |
| Elite education | 1395 | .682 | 1.823 | 0 | 0 | 0 | 0 | 6 |

Strct_pwr is the structural power of a CEO (duality + only insider + board committees); Own_pwr is the ownership power (1 if the percentage of a CEO's stock ownership or the percentage of a CEO's family members' ownership is higher than or equal to the sample median, or 0 otherwise); Expt_pwr is the expert power (1 if the year of experience as a CEO is higher than or equal to the sample median or if a CEO possesses a professional degree (CA or CMA or CFA), or 0 if otherwise; Prtge_pwr is the prestige power (CEO's other corporate board directorship + non-profit board directorship + elite education); Polit_pwr is the political power (1 if the CEO is or was a minister or member of parliament or government official or close relative to a politician or top post holder in the ruling party, or 0 if otherwise); Firm_siz refers to the natural log of total assets; Levge is measured as the ratio of total debt to total assets; RoA refers to the return on assets; AgE is the natural log of a firm's age; NegEarn is the indicator of net loss; Big 4 = 1 if a Big 4 audit firm audits the company; Growth_Rev is the percentage increase in sales revenue; Bd_siz represents the number of board members in the company; Bd_Ind represents the proportion of independent director; Inst_Own is the measure of the proportion of shares held by institutions.

CEO duality is 1 if the board president and CEO are the same person, or 0 if otherwise; CEO only insider is 1 if the CEO is the only person who represents top management in board meetings, and 0 if otherwise; CEO's board committees is 1 if a CEO's presence in corporate board committees is above or the same as the sample median, and 0 if otherwise.

A CEO's stock ownership represents the proportion of the company's shares that he or she owns. CEO family ownership represents the proportion of a company's shares held by the CEO's family members, less the CEO's stake.

The tenure of a CEO is measured in years of service. CEO Professional Degree = 1 if a CEO holds a CA, CMA, or a CFA certificate, and 0 if otherwise.

A corporate board directorship is the number of directorships of other corporate boards held by a CEO. The corporate board includes both private and public limited firms. The nonprofit board directorship is the total number of corporate nonprofit boards that a CEO is a member of. Elite education gets a score of 6 (most prestigious) if a CEO graduated from a university that was ranked within 1–50; 5 if the university was ranked within 51–100; 4 if the university was ranked within 101–150; 3 if the university was ranked within 151–200; 2 if the university was ranked within 201–250; 1 (least prestigious) if the university was ranked within 251–300; or 0 if otherwise. The university ranking is based on the Times Higher Education world university overall ranking.

and 25% for political power. 21% of the observations were audited by a Big 4 auditor, which is higher than that found by Muttakin, Khan, and Azim [124], using data from 2005 to 2009 in the Bangladesh context. They stated that a Big 4 audit firm had conducted an audit of 13.5% of the sample firms. This percentage is also higher than that found by Muttakin, Khan, and Mihret [62] using data for the period 2005 to 2013, for listed non-financial firms in Bangladesh. They stated that Big 4 audit firms had performed an audit on 17% of the sample companies. However, compared to other developing nations, the market share of Big 4 audit firms or their affiliates is much lower in Bangladesh. For example, Chihua, Tseng, and Chen [22] mentioned that on average, 73% of Taiwanese firms have been audited by a Big 4 auditor, whilst this portion is 62% for listed firms in Pakistan [125]. Khan, Muttakin, and Siddiqui [126] identified two main reasons for this. First, there is a low demand for high-quality audit service in family-dominated listed firms in Bangladesh. Second, these firms also offer low audit fees. Two of our main accrual earnings quality proxies (DisTAC and DisTA) showed that firms on average adjust accruals of 9.8% and 15.7% of total assets for DisTAC and DisTA respectively. Thus, the reported earnings are higher than they actually are due to the earnings management practices. Compared to other developed and developing economies the earnings quality of Bangladeshi listed firms is 2 times below that of U.S. companies [50] and 4% lower than that of Taiwanese listed firms [23].

Table 3 displays the correlation matrix for the independent and dependent variables. The correlation among the individual dimensions of CEO power is very weak, which indicates their distinctiveness and removes the concern of multicollinearity issues for this study.

## Multivariate analysis

**Earnings quality and dimensions of CEO power.** To verify Hypotheses 1, 2, 3, 4, and 5, Model 1 is estimated. Table 4 presents the outcomes of regressing the hypothesized variables onto earnings quality proxies. These results highlight some important insights. First, the statistically significant coefficients of the variables Lag DisTAC and Lag DisTA, indicate that there is a correlation between past and present values of earnings quality proxies. This provides the support of our choice of a dynamic estimation method, two-step system GMM, for this study.

Second, Hypothesis 1 states that CEOs with greater structural power are adversely related to accrual earnings quality. The estimation outcomes of Model 1 in Table 4 demonstrate that Strct_pwr ($\alpha_1$ = -0.039, p-value = 0.049; and $\alpha_1$ = -0.056, p-value = 0.048) has an inverse and statistically significant effect on both the earnings quality proxies DisTAC and DisTA, respectively, which supports Hypothesis 1. These findings imply that the existence of CEOs with high

**Table 3. Correlation matrix.**

| Variables | Strct_pwr | Own_pwr | Expt_pwr | Prtge_Pwr | Polit_pwr | DisTAC | DisTA | DisREV | Ab_P_C |
|---|---|---|---|---|---|---|---|---|---|
| Strct_pwr | 1.000 | | | | | | | | |
| Own_pwr | 0.095* | 1.000 | | | | | | | |
| Expt_pwr | 0.080* | 0.265* | 1.000 | | | | | | |
| Prtge_Pwr | -0.007 | 0.156* | 0.063* | 1.000 | | | | | |
| Polit_pwr | 0.113* | -0.068* | -0.178* | 0.016 | 1.000 | | | | |
| DisTAC | 0.001 | 0.068* | 0.019 | 0.003 | -0.077* | 1.000 | | | |
| DisTA | -0.005 | 0.092* | 0.015 | -0.022 | -0.008 | 0.428* | 1.000 | | |
| DisREV | 0.032 | 0.032 | 0.035 | -0.039 | -0.029 | 0.508* | 0.680* | 1.000 | |
| Ab_P_C | -0.051* | -0.090* | -0.056* | 0.057* | -0.010 | -0.444* | -0.584* | -0.660* | 1.000 |

* indicates significance at 10% level

**Table 4. Dimensions of CEO power and earnings quality: System GMM estimates.**

| Variables | Pred Sign | DisTAC | P-value | DisTA | P-value |
|---|---|---|---|---|---|
| Lag DisTAC | | 0.226*** | 0.002 | | |
| Lag DisTA | | | | 0.338*** | 0.001 |
| Strct_pwr | - | -0.039** | 0.049 | -0.056** | 0.048 |
| Own_pwr | - | -0.013 | 0.618 | -0.005 | 0.921 |
| Expt_pwr | - | -0.037*** | 0.007 | -0.056** | 0.043 |
| Prtge_Pwr | - | -0.001 | 0.959 | 0.003 | 0.718 |
| Polit_pwr | - | -0.049** | 0.041 | -0.189 | 0.038 |
| Firm_siz | | 0.004 | 0.247 | 0.011** | 0.030 |
| Bd_siz | | 0.001 | 0.856 | 0.003 | 0.790 |
| Bd_Ind | | 0.011 | 0.347 | 0.016 | 0.504 |
| Inst_Own | | -0.001 | 0.750 | 0.001 | 0.649 |
| Levge | | -0.071 | 0.242 | -0.126 | 0.187 |
| RoA | | -0.739*** | (0.008) | -0.810 | 0.219 |
| AgE | | 0.006 | 0.708 | -0.037 | 0.296 |
| NegEarn | | -0.097** | 0.028 | -0.105 | 0.194 |
| Big4 | | -0.011 | 0.667 | 0.034 | 0.607 |
| Growth_Rev | | -0.001 | 0.871 | 0.001 | 0.642 |
| Year and Ind. Dum. | | Included | | Included | |
| No. of firm-years | | 1034 | | 1034 | |
| F-stat | | 25.05*** | 0.001 | 14.51*** | 0.001 |
| Number of firms considered | | 165 | | 165 | |
| Number of instruments used | | 122 | | 122 | |
| A.R. (1) p-value | | | 0.001 | | 0.012 |
| A.R. (2) p-value | | | 0.754 | | 0.249 |
| Hansen *J* stats p-value | | | 0.705 | | 0.492 |

GMM estimation with Windmeijer [31] corrected standard errors. Significant values of 1%, 5%, and 10% are indicated by the symbols ***, **, and *, respectively.

structural power is connected to the reduction in earnings quality by 3.9% and 5.6% of total assets for the accrual quality proxies DisTAC and DisTA, respectively. These results also imply that structurally powerful CEOs in Bangladesh manage accruals to show better firm performance. These findings concur with prior studies by Koo and Kim [5], Mande and Son [7], Nuanpradit [18], Bouaziz, Salhi, and Jarboui [61], Shiah-Hou [17] and Muttakin, Khan, and Mihret [62].

These results are also consistent with the arguments by prominent agency theorists that excessive formal power promotes CEO entrenchment, which in turn leads to decisions that are contrary to the interests of minority shareholders, and ultimately they lose their wealth [2, 4, 127]. Ahmed and Uddin [63] mentioned that CEOs in Bangladesh have enormous power, and other top executives are prone to follw the instructions of CEOs even at the expense of general shareholders. This is because the recruitments, promotions, bonuses and fireing from the job are not performed in a formal way. These tasks are performed informally, arbitarily, and with direct intervesions by CEOs [63].

Hypothesis 2 predicts that CEOs with inordinate ownership power are harmful to earnings quality. The empirical findings displayed in Table 4 attest that ownership power has an insignificant bearing on either earnings quality proxy, which refutes Hypothesis 2. These results are consistent with Le, Kweh, Ting, and Nourani [23], Hribar and Nichols [72] and Rashid [73]. These results imply that CEOs with significant ownership influence are not linked with the

decrease in earnings quality. There could be at least three explanations for these findings. First, some scholars argue that a CEO may hold different dynamics of power, but it is not necessary to use all the bases of power to achieve any objective [128, 129]. Second, unlike CEOs with little or no equity ownership, the compensation and employment contracts of CEOs with high ownership power are not sensitive to the firm's accounting performance [1]. As a result, they do not need to mask poor financial performance through earnings management techniques [1]. Third, the identity and goodwill of CEOs with high ownership stakes are associated with their firms [130]. Thus, they have the fear of losing their reputation if they engage in any accounting manipulation.

Hypothesis 3 states that CEOs with greater expertise will be tied up with lower earnings quality. The statistically significant negative coefficients of Expt_Pwr ($\alpha_3$ = -0.037, p-value = 0.007; and $\alpha_3$ = -0.056, p-value = 0.043) across both earnings-quality proxies support Hypothesis 3. These results suggest that the presence of more expert CEOs are associated with 3.7% increases in discretionary current accruals (DisTAC), and 5.6% increase of abnormal accruals (DisTA). In other words, CEOs who hold their position for a longer tenure or have a professional accounting degree are more involved in earnings manipulation activities in Bangladesh. The findings of this research are consistent with Shiah-Hou [17], Hu and Gan [19], Le, Kweh, Ting, and Nourani [23], Chihua, Tseng, and Chen [22], Altunbaş, Thornton, and Uymaz [75], and Priscilla and Siregar [86]. These results are meaningful in the context of Bangladesh, as in many cases, expert CEOs both manage and control corporate decisions and take actions that debilitate the wealth of minority shareholders [63]. The findings are also consistent with agency theory propositions that managerial expertise augments managerial power, which motivates CEOs to apply more earnings management techniques to inflate earnings and keep secret their ingrained behaviors [22, 40, 131].

Hypothesis 4 predicts that CEOs with greater prestige power will be adversely attached to the quality of accrual income. The results in Table 4 illustrate that the impact of prestige power is insignificant for both proxies, which rejects Hypothesis 4. One potential explanation for the insignificant impact is that CEOs with high prestige power do not engage in unethical practices to protect their reputation and to increase their social status.

Hypothesis 5 states that politically connected CEOs maintain lower quality earnings. The significant and negative coefficients of Polit_pwr ($\alpha_5$ = -0.049, p-value = 0.041; and $\alpha_5$ = -0.189, p-value = 0.038) across both earnings-quality proxies support Hypothesis 5. These findings offer evidence that politically connected CEOs are responsible for a 4.9% increment in discretionary current accruals (DisTAC), and an 18.9% surge in abnormal total accruals (DisTA) in listed non-financial firms in Bangladesh.

These results are logical for at least two reasons. First, although politically connected CEOs make a huge amount of profit, they are not concerned about depriving non-controlling shareholders of their financial rights by manipulating financial statements. They transfer the resources of a cash cow firm to their different private projects and declare and pay very insignificant dividends [82]. Second, regulatory bodies play a silent role regarding the wrongdoing of politically connected CEOs. This is because if they take any action against these politically connected CEOs, they may lose their position or even be shifted to other less important sections of the government [63]. Interestingly, the top positions of corporate regulatory bodies are also filled by explicitly politically affiliated people connected to the ruling party. As a result, these top officials of regulatory bodies offer preferential treatment to politically connected CEOs as they follow the same political ideology.

Results related to control variables show that RoA and NegEarn are negatively associated with the accrual quality proxies, suggesting that to show better firm performance and to hide actual performance, firms in Bangladesh compromise with earnings quality. Also, the variable

Levge is conversely linked with earnings quality, suggesting that with the rise in firms' debt, earnings quality also reduces. This makes sense given that the Big 4 audit companies do not significantly affect the quality of earnings in the Bangladesh setting. However, Firm_siz shows a significantly positive association with earnings quality, indicating that larger firms maintain better quality earnings. Overall, these findings endorse the agency theory stance as powerful CEOs are detrimental to a firm's earnings quality, especially if they possess excessive structural, expert, and political power.

Finally, the post estimation results of the system-GMM estimates disclose that our estimated models are properly determined as the F-stats are statistically significant at 1% level for both earnings quality proxies. The three main diagnostic results are as follows. First, the probability values of A.R. (1) are statistically significant indicating that there is a high first-order autocorrelation in each estimation; however, the probability values for A.R. (2) are insignificant, showing proof of no second-order autocorrelation. Next, the Hansen J-statistics report insignificant p-values, which certify the validity of instruments used in this study. Finally, this study used a lower number of instruments as against the number of firms, which keeps these estimates free of the instrument proliferation problem [132].

## Tests of robustness

**Alternate earnings quality proxy.**　For assessing the robustness of our main results, we applied an alternate earnings quality proxy, discretionary revenue, as the dependent variable, and estimates Model 1. The proxy is extracted from revenue-accrual quality, as proposed by Stubben [133] and adopted from Hope, Thomas, and Vyas [50]. In this measure, receivables accrual is modeled as a function of change in revenues. This study, following prior studies, estimated the following equation cross-sectionally for each year and every industry, with a minimum of 8 firm-year observations to confirm enough data to capture the estimated parameters [104, 106, 107]. Each of the constituents in this equation is scaled by the lag total assets.

$$\Delta AcR_{i,t} = \delta_0 + \delta_1 \Delta Rev_{i,t} + \in_{i,t} \tag{4}$$

In this equation, $\Delta AcR_{i,t}$ stands for yearly variation in accounts receivable, $\Delta Rev_{i,t}$ depicts yearly change in sales revenues. Discretionary revenues (DisREV) are the residuals' absolute values from Eq 4. Following Hope, Thomas, and Vyas [50], this study multiplied these absolute values by -1, so the bigger values of DisREV, the better the quality of earnings.

The findings in Table 5 reveal that structural power (Strct_pwr), expert power (Expt_pwr), and political power (Polit_pwr) have statistically significant and negative coefficients at $\alpha_1 =$ -0.020, p-value = 0.070; $\alpha_3 =$ -0.030, p-value = 0.046; and $\alpha_5 =$ -0.063, p-value = 0.063, respectively, for discretionary revenue (DisREV). However, the association between the quality of earnings and the ownership and prestige power of CEOs is statistically insignificant. These results also support Hypotheses 1, 3, and 5, and suggest that CEOs with substantial structural, expert, and political power are associated with sales manipulation that lower the quality of accounting earnings.

**Additional analysis.**　This research employed a real activity management proxy, abnormal production costs (Ab_P_C), to examine whether powerful CEOs inflate earnings using real transaction management techniques besides accrual earnings management. Under this strategy, businesses create more units than necessary to reduce their cost per unit or cost of products sold, which raises their gross profit margin [134–136]. The production cost manipulation approach is primarily used by manufacturing companies [134], even though Roychowdhury [135] listed three unique methods of real activity management: discretionary spending manipulation, cash flow manipulation, and production cost manipulation. This study tests only

**Table 5. CEO power aspects and earnings quality (DisREV).**

| Variables | Pred. Sign | DisREV | P- value |
|---|---|---|---|
| Lag DisREV | | 0.231* | (0.001) |
| Strct_pwr | - | -0.020* | (0.070) |
| Own_pwr | - | 0.015 | (0.232) |
| Expt_pwr | - | -0.030** | (0.046) |
| Prtge_Pwr | - | -0.003 | (0.566) |
| Polit_pwr | - | -0.063* | (0.063) |
| Firm_siz | | 0.007* | (0.072) |
| Bd_siz | | 0.002 | (0.619) |
| Bd_Ind | | 0.015** | (0.013) |
| Inst_Own | | 0.001 | (0.493) |
| Levge | | -0.011 | (0.724) |
| RoA | | -0.527*** | (0.004) |
| AgE | | -0.046** | (0.034) |
| NegEarn | | -0.022 | (0.254) |
| Big4 | | 0.022 | (0.401) |
| Growth_Rev | | 0.001 | (0.996) |
| Year and Ind. Dum. | | Included | |
| No. of firm-years | | 1034 | |
| F-stat | | 6.60*** | 0.001 |
| Number of firms considered | | 165 | |
| Number of instruments used | | 122 | |
| A.R. (1) p-value | | | 0.010 |
| A.R. (2) p-value | | | 0.673 |
| Hansen $J$ stats p-value | | | 0.652 |

GMM estimation with Windmeijer [31] corrected standard errors. Significant values of 1%, 5%, and 10% are indicated by the symbols ***, **, and *, respectively.

production cost manipulation for additional analysis purposes, as 88% of our sample firms are manufacturing firms. Thus, the probability is higher that the CEOs will prefer production cost manipulation to discretionary expense manipulation or cash flow manipulation for the management of real earnings. This proxy is measured using Roychowdhury's [135] model. Specifically, Eq 5 is estimated to get the proxy values:

$$\frac{Prod_{it}}{Asst_{it-1}} = \gamma_0 + \gamma_1 \left( \frac{1}{Asst_{it-1}} \right) + \gamma_2 \left( \frac{Sls_{it}}{Asst_{it-1}} \right) + \gamma_3 \left( \frac{\Delta Sls_{it}}{Asst_{it-1}} \right) + \gamma_4 \left( \frac{\Delta Sls_{it-1}}{Asst_{it-1}} \right) + \varepsilon_{it} \quad (5)$$

Where $Prod_{it}$ represents the total cost of the products or services that were sold out plus the annual change in the value of the inventory. $Asst_{it-1}$ represents worth of assets at the outset of a year. $Sls_{it}$ is for sales during a year, $\Delta Sls_{it}$ represents yearly change in sales amount, and $\Delta Sls_{it-1}$ represents one year lagged change in sales value. Abnormal production costs (Ab_P_C) are the residuals from Eq (5). This study then estimates Model 1 to ascertain the consequence of excessive CEO power upon abnormal production costs. This study expects that CEOs' power dimensions and abnormal production costs will be positively associated.

The results in Table 6 show that structural power (Strct_pwr), expert power (Expt_pwr), and political power (Polit_pwr) have statistically significant and positive coefficients at $\alpha_1 =$ 0.209, p-value = 0.026; $\alpha_3 = 0.188$, p-value = 0.010; and $\alpha_5 = 0.322$, p-value = 0.032, respectively, for abnormal production costs (Ab_P_C). However, ownership and prestige power

**Table 6. CEO power facets and earnings quality (Ab_P_C).**

| Variables | Pred. Sign | Ab_P_C | P-value |
|---|---|---|---|
| Lag Ab_P_C | | 0.279* | (0.091) |
| Strct_pwr | + | 0.209** | (0.026) |
| Own_pwr | + | -0.157 | (0.209) |
| Expt_pwr | + | 0.188** | (0.010) |
| Prtge_Pwr | + | 0.012 | (0.693) |
| Polit_pwr | + | 0.322** | (0.032) |
| Firm_siz | | -0.047** | (0.022) |
| Bd_siz | | 0.016 | (0.354) |
| Bd_Ind | | -0.029 | (0.527) |
| Inst_Own | | 0.001 | (0.942) |
| Levge | | 0.318** | (0.016) |
| RoA | | 0.627*** | (0.001) |
| AgE | | 0.106 | (0.166) |
| NegEarn | | 0.231 | (0.113) |
| Big 4 | | -0.116 | (0.272) |
| Growth_Rev | | -0.001 | (0.535) |
| Year and Ind. Dum. | | Included | |
| No. of firm-years | | 869 | |
| F-stat | | 71.37*** | 0.001 |
| Number of firms considered | | 161 | |
| Number of instruments used | | 109 | |
| A.R. (1) p-value | | | 0.041 |
| A.R. (2) p-value | | | 0.299 |
| Hansen J stats p-value | | | 0.763 |

GMM estimation with Windmeijer [31] corrected standard errors. Significant values of 1%, 5%, and 10% are indicated by the symbols ***, **, and *, respectively.

show an insignificant effect on Ab_P_C. These results support hypotheses 1, 3, and 5, and suggest that CEOs with high structural, expert, and political power use the production cost manipulation technique to inflate the reported earnings.

This study also investigates whether the effect of powerful CEOs on earnings quality varies between large and small firms in Bangladesh. Prior studies suggest that small firms generally face more liquidity constraints than large firms, as these firms have less access to commercial paper and public debt [137–141]. Accordingly, small firms depend on banks and other financial institutions to arrange a significant part of their required funds [138], whereas larger companies are less financially constrained and have fewer issues with money when investing in a project [142]. In addition, small firms face more idiosyncratic risks than large firms [137]. Thus, the behavior of powerful CEOs concerning earnings quality may differ in large and small firms. This study argues that as the scope of fund generation for small firms is relatively narrow, the CEOs of these firms face more pressure to satisfy the debt covenants of banks and financial institutions. These CEOs may engage in more earnings management practices to reduce the perceived risks of creditors (i.e., the probability of not paying the debt) and to raise additional funds under favorable credit terms [121]. Accordingly, this study predicts that– powerful CEOs in small firms are associated with more earnings management practices than powerful CEOs in big firms.

To investigate this issue, this study divides the sample into two groups (large and small firms), taking the average firm size, 21.67, as the cutoff point. Firm-year observations with a

**Table 7. CEO power dynamics and earnings quality: Large versus small firms.**

| Variables | Large Firms | | Small Firms | | Large Firms | | Small Firms | |
|---|---|---|---|---|---|---|---|---|
| | DisTAC | P-val | DisTAC | P-val. | DisTA | P-val. | DisTA | P-val. |
| L. DisTAC | 0.179** | 0.049 | 0.288*** | 0.001 | | | | |
| L. DisTA | | | | | 0.281*** | 0.001 | 0.258** | 0.028 |
| Strct_pwr | -0.060** | 0.044 | -0.034** | 0.045 | -0.090** | 0.022 | -0.070** | 0.047 |
| Own_pwr | 0.060** | 0.016 | -0.013 | 0.461 | 0.049 | 0.439 | -0.032 | 0.417 |
| Expt_pwr | -0.036 | 0.155 | -0.024 | 0.248 | -0.032 | 0.338 | 0.013 | 0.730 |
| Prtge_Pwr | -0.001 | 0.862 | 0.006 | 0.417 | 0.015 | 0.234 | -0.005 | 0.639 |
| Polit_pwr | -0.035 | 0.310 | -0.024 | 0.511 | -0.137 | 0.178 | -0.029 | 0.421 |
| Firm_siz | 0.002 | 0.707 | 0.001 | 0.718 | 0.010 | 0.235 | 0.014*** | 0.007 |
| Levge | -0.153*** | 0.010 | -0.036 | 0.504 | -0.371* | 0.082 | -0.051 | 0.633 |
| RoA | -0.654** | 0.013 | 0.044 | 0.747 | -0.795 | 0.272 | -0.500 | 0.171 |
| AgE | 0.030* | 0.100 | -0.020 | 0.483 | 0.008 | 0.854 | -0.087* | 0.069 |
| NegEarn | -0.070* | 0.050 | 0.003 | 0.946 | -0.112 | 0.327 | -0.046 | 0.473 |
| Big4 | 0.014 | 0.523 | -0.007 | 0.790 | -0.006 | 0.900 | 0.092 | 0.161 |
| Growth_Re | 0.001 | 0.497 | 0.001 | 0.432 | -0.001 | 0.540 | 0.001 | 0.173 |
| B_size | 0.001 | 0.984 | 0.002 | 0.769 | 0.005 | 0.738 | -0.004 | 0.639 |
| B_Ind | -0.001 | 0.970 | 0.026 | 0.107 | 0.032 | 0.240 | 0.056* | 0.096 |
| Inst_Own | -0.001 | 0.657 | 0.001 | 0.626 | -0.002* | 0.096 | 0.001 | 0.386 |
| F-stat | 17.94*** | 0.001 | 33.31*** | 0.001 | 15.35*** | 0.001 | 17.14*** | 0.001 |
| Year & Ind. Dum. | Yes | | Yes | | Yes | | Yes | |
| Observations | 520 | | 514 | | 520 | | 514 | |
| A.R. (1) p-value | 0.006 | | 0.001 | | 0.048 | | 0.002 | |
| A.R. (2) p-value | 0.125 | | 0.989 | | 0.468 | | 0.508 | |
| Hansen J stat p-val. | 0.891 | | 0.960 | | 0.852 | | 0.980 | |

GMM estimation with Windmeijer [31] corrected standard errors. Significant values of 1%, 5%, and 10% are indicated by the symbols ***, **, and *, respectively.

mean firm size (log of total assets) of less than 21.67 are classified as small firms, and firm-year observations with a mean firm size equal to or greater than 21.67 are classified as large firms. Then, this study estimates Model 1 for both large and small firms, considering the two earnings quality proxies, DisTAC and DisTA. The estimated results are presented in Table 7.

The results in Table 7 show that structural power (Strct_pwr) has negative and statistically significant coefficients at the 5% level for both large and small firms, and across both the earnings quality proxies, DisTAC and DisTA. Expert power (Expt_pwr) has negative coefficients for both large and small firms for the proxy DisTAC. For the proxy DisTA, expert power has a negative coefficient for large firms and a positive coefficient for small firms. The coefficients of expert power are statistically insignificant for large and small firms altogether. Political power has negative coefficients for both large and small firms and across both the earnings quality proxies, DisTAC and DisTA. The coefficients of political power are also statistically insignificant. The rest of the two power dynamics, prestige (Prtge_Pwr) and ownership power (Own_pwr), also suggest an insignificant difference between large and small firms, except for a positive and statistically significant association between ownership power and the earnings quality proxy, DisTAC, for large firms. Thus, overall, these findings imply that, in the context of Bangladesh, the effect of powerful CEOs on earnings quality for large and small firms is identical. One important reason for the insignificant difference between large and small firms to the influence of powerful CEOs is that the capital market of Bangladesh is small and the difference between the listed non-financial firms in terms of firm size or total assets is

insignificant. As can be seen from the descriptive statistics, the standard deviation of firm size is very low at 1.52. Also, the mean, median, and third quartile values are 21.67, 21.65, and 22.58, respectively, which are very close to each other.

## Discussion and conclusion

This research expands the body of knowledge by offering empirical evidence regarding the impact of a CEO's structural, ownership, expert, prestige, and political power on earnings quality in an emerging economy context. Using accrual earnings quality proxies, real earnings management measures, and two-step system GMM, our findings demonstrate that CEOs with more structural power, considerable expertise, and politically connected CEOs are involved in inferior financial reporting practices. These CEOs, besides accrual earnings management, apply real activity management techniques to manipulate the reported earnings. However, CEOs with high ownership and prestige power have an insignificant effect on earnings quality. These results suggest that powerful CEOs, especially those with excessive structural and expert power and politically connected CEOs are detrimental to the interests of residual claimants (specifically minority stockholders) in emerging economies like Bangladesh. Further investigation demonstrates that in the context of Bangladesh, there are no appreciable differences between large and small listed firms in terms of the impact of CEO power on earnings quality.

The results of this study have some crucial policy suggestions. This study offers two policies. First, a restriction on CEOs' direct political affiliation may improve the corporate regulatory environment and, ultimately, the reported earnings quality. This will reduce CEOs' undue pressure over regulatory bodies. Second, a limit on CEOs' tenure and the involvement of one or two top executives other than the CEO on the corporate board may significantly improve internal corporate governance. Their consequent involvement in strategic decision-making will minimize CEOs' autocracy.

This study has two main limitations. First, for earnings quality measures, this study used only accounting-based accrual earnings quality proxies and one real earnings management proxy. Second, as this study did not find appropriate external instruments, this study used only internal instruments to address the issues of endogeneity. Future studies may examine the impact of various CEO power facets on cash flow manipulation, discretionary expense manipulation, and the market-based earnings quality measures. Future research may also investigate whether the effects of powerful CEOs on earnings quality differ between family-controlled and non-family entities. Finally, future research may focus on using external exogenous instruments to address the endogeneity in CEO power dimensions.

## Author Contributions

**Conceptualization:** Mohd Zulkhairi Mustapha, Azlina Abdul Jalil.

**Formal analysis:** H. M. Arif.

**Investigation:** H. M. Arif.

**Methodology:** H. M. Arif, Azlina Abdul Jalil.

**Project administration:** Mohd Zulkhairi Mustapha.

**Supervision:** Mohd Zulkhairi Mustapha, Azlina Abdul Jalil.

**Writing – original draft:** H. M. Arif.

**Writing – review & editing:** Mohd Zulkhairi Mustapha, Azlina Abdul Jalil.

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
