## [Decision Letter · Decision Letter 0]

7 Jun 2022

PONE-D-22-04720Do powerful CEOs matter for earnings quality? Evidence from BangladeshPLOS ONE

Dear Dr. Mustapha,

Thank you for submitting your manuscript to PLOS ONE. After careful consideration, we feel that it has merit but does not fully meet PLOS ONE’s publication criteria as it currently stands. Therefore, we invite you to submit a revised version of the manuscript that addresses the points raised during the review process.

The manuscript requires further improvements with reference to the theoretical part, as well as the quantitative approach. Also, the English language should be refined.

We look forward to receiving your revised manuscript.

Kind regards,

Stefan Cristian Gherghina, PhD. Habil.

Academic Editor

PLOS ONE

**Journal requirements:**

 2. PLOS ONE does not copy edit accepted manuscripts (https://journals.plos.org/plosone/s/criteria-for-publication#loc-5). To that effect, please ensure that your submission is free of typos and grammatical errors.

*For this observational study, please avoid causal-sounding language (such as 'impact' or 'effect') when reporting associations."

“Unfunded study”

d) If you did not receive any funding for this study, please state: “The authors received no specific funding for this work.

“NO authors have competing interests”

Reviewers' comments:

Reviewer's Responses to Questions

**Comments to the Author**

1. Is the manuscript technically sound, and do the data support the conclusions?

Reviewer #1: Yes

Reviewer #2: No

Reviewer #3: Yes

Reviewer #4: Yes

2. Has the statistical analysis been performed appropriately and rigorously? 

Reviewer #1: Yes

Reviewer #2: No

Reviewer #3: Yes

Reviewer #4: Yes

3. Have the authors made all data underlying the findings in their manuscript fully available?

Reviewer #1: Yes

Reviewer #2: No

Reviewer #3: Yes

Reviewer #4: No

4. Is the manuscript presented in an intelligible fashion and written in standard English?

Reviewer #1: Yes

Reviewer #2: No

Reviewer #3: Yes

Reviewer #4: Yes

5. Review Comments to the Author

Reviewer #1: Some amendment need to be done in sections below:

1. Literature review- should include discussion on earnings quality

2 Research approach- need to revise and revisit based on following items.

i. Explanation for excluding for firm less than 8 entities

ii No of firms 165 for 10 years should be 1650 not 1395

iii. Why 8 observations? explain

iv. score of 1 if it is above or same as the sample median - confusing statement.

v. CEO expert = experience years OR professional degree or BOTH

vi. Corporate board membership is it refer to CEO multiple directorships?

vii. Improve the measurement of model 1 variables.

3 Results- check for the result- figure in table should consistent with text.

i. suggest to provide the descriptive table based on the raw data before coded it to 0,1 till 8.

ii. Suggest to place a legend for each of variables mentioned in Table 2

iii. Suggest may also include DV in correlation table as to determine the strength of relationship between DV and IVs .

iv. May provide an explanation/ discussion/ justification of result for relationship between CEO power & EQ not only mention the statistical results.

v. Need to justify as CEO ownership is on average 77% but the result is insignificant

vi. Suggest to exclude predicted sign for control variables in Table 4, 5 & 6

vii. Additional analysis-

Justify why only production cost as Roychowdhury (2016) focused on 3 elements of REM which is CFO, production & discretionary expenses.

Table 6- Explain why only 161 firms not 165.

4. Discussion

Further, the lack of skilled, competent, and independent manpower in regulatory bodies is also a prime cause for the poor monitoring system- A statement based on previous study- any reference? it is not tested in this study.

Reviewer #2: I have comments to offer.I have comments to offer.I have comments to offer.I have comments to offer.I have comments to offer.I have comments to offer.I have comments to offer.I have comments to offer.

Reviewer #3: I found that the authors put their efforts into investigating effects of powerful Chief Executive Officers (CEOs) on earnings quality in a setting where CEOs are so dominating, they exercise their influence over corporate regulatory bodies. Using a 10-year longitudinal data for the period 2010 to 2019 and 1,395 firm-year observations from listed non-financial firms in Bangladesh, they found that CEOs political power and CEOs with high structural and expert power have a significant detrimental effect on earnings quality. Whereas ownership and prestige power have an insignificant impact on earnings quality. These powerful CEOs use accrual and real activities manipulation techniques together to manage the earnings. I found that this study addresses the new issue and it will be really interesting for readers. This study is overall good and have a good format. The concept of understanding the role of CEO’s power on earnings quality will create future scope of this study. Authors have specified the problems in a correct manner and written the results in detailed manner. Model seems fine to me but hypotheses need modifications. Further, I have a few major concerns before this paper can be consider for publication in PLOS One.

1. First, if authors want to make any additional analysis then they can make sub-sample of data to better deal with the issue of financing constraints. Gertler and Gilchrest (1993) state that smaller firms have more external finance premium than larger firms, which could be due to two reasons: first, large firms have more collateral assets that help them to easily finance their investments and secondly, large firms might be having their business group that helps them to use their own internal capital market. Gertler and Gilchrist (1994) argued that small companies act as a proxy for financially constrained firms because these companies exhibit greater bank dependencies, cannot issue public debt and face a higher level of idiosyncratic risk. Apart from this, smaller firms are usually younger, with a high level of firm-specific risk and have less collateral, thereby reducing the possibility of attracting external finance. Gupta et el. (2021) also documented that small firms are more financially constrained than large firms. Here, authors should divide the sample on the basis of firm size (small firms and large firms) to know the how CEO’s Power affect firm earning quality for small firms and large firms as small firms are different than large firms. Authors should also cite the following studies while discussing about financing constraints in terms of firm size.

a. Gertler, M. and Gilchrist, S. (1993), “The role of credit market imperfections in the monetary transmission mechanism: arguments and evidence”, The Scandinavian Journal of Economics, Vol. 95 No. 1, pp. 43-64.

b. Gertler, M. and Gilchrist, S. (1994), “Monetary policy, business cycles, and the behavior of small manufacturing firms”, Quarterly Journal of Economics, Vol. 109 No. 2, pp. 309-340.

c. Gupta, G., Mahakud, J. and Verma, V. (2021), "CEO's education and investment–cash flow sensitivity: an empirical investigation", International Journal of Managerial Finance, Vol. 17 No. 4, pp. 589-618.

2. Authors need to write additional hypothesis for small and large size firms analysis.

3. Results of the empirical analysis are clearly presented. But after the implemented changes mentioned for the financial constraints, it should be re-written accordingly.

4. Authors should also include the following studies of CEO’s characteristics to strengthen their literature review.

1. Gupta, G., 2022. CEO's age and investment‐cash flow sensitivity. Managerial and Decision Economics.

2. Farag, H., & Mallin, C. (2018). The influence of CEO demographic characteristics on corporate risk-taking: Evidence from Chinese IPOs. The European Journal of Finance, 24, 1528–1551.

3. Gupta, G. (2021). CEO's educational background, economic policy uncertainty and investment-cash flow sensitivity: Evidence from India. Applied Economics, 1–12.

4. Naheed, R., Jawad, M., Naz, M., Sarwar, B., & Naheed, R. (2021). Managerial ability and investment decisions: Evidence from Chinese market. Managerial and Decision Economics, 42, 985–997.

Reviewer #4: This study investigates the effects of powerful Chief Executive Officers (CEOs) on earnings quality in a setting where CEOs are so dominating, they exercise their influence over corporate regulatory bodies. Using a 10-year longitudinal data for the period 2010 to 2019 and 1,395 firm-year observations from listed non-financial firms in Bangladesh, we found that CEOs political power and CEOs with high structural and expert power have a significant detrimental effect on earnings quality.

After carefully reading through your paper, I am happy to report that it is highly original, well-conceived, designed, analysed, concluded and written with significant potential to contribute to the existing literature. Indeed, I enjoyed reading it and therefore, I applaud the author/s for the excellent effort in writing this paper.

I have the following minor suggestions, which can help improve the paper further.

Major issues:

You need to elaborate on research gaps further and then link those gaps to the contributions in results which will help you motivate the research context.

Literature review and hypotheses Development - please enhance your hypotheses by: (i) drawing on the theory; (ii) empirical literature; (iii) research setting/contextual insights; and (iv) then setting up your hypotheses. You will do this for each hypothesis. You can do so by drawing on both seminal (old) and recently (newly) published studies.

Methodology- You need to add other corporate governance variables as control variables such as board size, board independence and ownership structure.

You need to explain the GMM approach further and show which instruments you use as exogenous variable(s) and why.

If you did not consider any instrumental variable(s), you should use 2SLS to make sure that your results are not driven by endogeneity issues.

In your descriptive, I can notice that you may use propensity score matching to sole self-selection bias.

You use earnings management proxies to test earnings quality, why? Why you did not use the earnings response coefficient!

Empirical findings - please link your findings more strongly to the: (i) theory, (ii) empirics, (iii) context; and (iv) highlight their economic, academic/research and policy implications. Closely link up and cite the papers that you have discussed in the background, theory and empirical literature review & and hypotheses development section to the findings you are presenting here.

Conclusion - Please outline a summary of findings, contributions, implications, limitations and avenues for future research. Especially, expand the discussions relating to implications, limitations and avenues for future research.

Minor issues

The standard of writing needs some work. A more careful reading and restructuring will help eliminate some remaining typos.

Please refer to some relevant literature from the journal and wider literature such as the below references.

6. PLOS authors have the option to publish the peer review history of their article (what does this mean?). If published, this will include your full peer review and any attached files.

Reviewer #1: No

Reviewer #2: No

Reviewer #3: No

Reviewer #4: No

---

## [Author Response · Author response to Decision Letter 0]

27 Aug 2022

Dear Reviewers,

We have addressed all the comments that you have made. We are very thankful for the comments. It has surely improved the paper.

---

## [Decision Letter · Decision Letter 1]

26 Sep 2022

PONE-D-22-04720R1Do powerful CEOs matter for earnings quality? Evidence from BangladeshPLOS ONE

Dear Dr. Mustapha,

Thank you for submitting your manuscript to PLOS ONE. After careful consideration, we feel that it has merit but does not fully meet PLOS ONE’s publication criteria as it currently stands. Therefore, we invite you to submit a revised version of the manuscript that addresses the points raised during the review process.

The author(s) should implement further the suggestions and recommendations of the second referee.

We look forward to receiving your revised manuscript.

Kind regards,

Stefan Cristian Gherghina, PhD. Habil.

Academic Editor

PLOS ONE

Journal Requirements:

Reviewers' comments:

Reviewer's Responses to Questions

**Comments to the Author**

1. If the authors have adequately addressed your comments raised in a previous round of review and you feel that this manuscript is now acceptable for publication, you may indicate that here to bypass the “Comments to the Author” section, enter your conflict of interest statement in the “Confidential to Editor” section, and submit your "Accept" recommendation.

Reviewer #2: All comments have been addressed

Reviewer #3: All comments have been addressed

Reviewer #4: All comments have been addressed

2. Is the manuscript technically sound, and do the data support the conclusions?

Reviewer #2: Yes

Reviewer #3: Yes

Reviewer #4: Yes

3. Has the statistical analysis been performed appropriately and rigorously? 

Reviewer #2: Yes

Reviewer #3: Yes

Reviewer #4: Yes

4. Have the authors made all data underlying the findings in their manuscript fully available?

Reviewer #2: Yes

Reviewer #3: No

Reviewer #4: Yes

5. Is the manuscript presented in an intelligible fashion and written in standard English?

Reviewer #2: Yes

Reviewer #3: Yes

Reviewer #4: Yes

6. Review Comments to the Author

Reviewer #2: PONE-D-22-04720R1

Do powerful CEOs matter for earnings quality? Evidence from Bangladesh

Comments fully addressed, no further comments.

Reviewer #3: Authors did not considered any of my comments. They have to acknowledge the reviewer comments to make it publishable.

Reviewer #4: Thanks for addressing all my comments efficiently. This study investigates the effects of powerful Chief Executive Officers (CEOs) on earnings quality in a setting where CEOs are so dominating, they exercise their influence over

corporate regulatory bodies. Using 10-year longitudinal data for the period from 2010 to 2019 and 1,395 firm-year observations from listed non-financial firms in Bangladesh, we found that CEOs’ political power and CEOs with high structural and expert power have a significant detrimental effect on earnings quality. Ownership and prestige power have an insignificant impact on earnings quality. These powerful CEOs use accrual and real activities manipulation techniques together to manage the earnings. This study uses the system generalized method of moment estimates for

estimation purpose, and the results remain robust when alternative earnings quality proxies are used. Taken together, our results suggest that CEOs’ political duality (i.e., serving simultaneously as a member of parliament and a CEO) should be restricted and that a CEO’s tenure should be limited to a reasonable period. This research adds to the existing body of knowledge by supplying empirical proof about the power dynamics of CEOs on earnings quality, including political and prestige power.

7. PLOS authors have the option to publish the peer review history of their article (what does this mean?). If published, this will include your full peer review and any attached files.

Reviewer #2: No

Reviewer #3: No

Reviewer #4: **Yes: **Ahmed A. Elamer

---

## [Author Response · Author response to Decision Letter 1]

11 Oct 2022

We have addressed all the comments made by the reviewers. Thank you

---

## [Decision Letter · Decision Letter 2]

18 Oct 2022

Do powerful CEOs matter for earnings quality? Evidence from Bangladesh

PONE-D-22-04720R2

Dear Dr. Mustapha,

We’re pleased to inform you that your manuscript has been judged scientifically suitable for publication and will be formally accepted for publication once it meets all outstanding technical requirements.

Kind regards,

Stefan Cristian Gherghina, PhD. Habil.

Academic Editor

PLOS ONE

Additional Editor Comments (optional):

Reviewers' comments:

Reviewer's Responses to Questions

**Comments to the Author**

1. If the authors have adequately addressed your comments raised in a previous round of review and you feel that this manuscript is now acceptable for publication, you may indicate that here to bypass the “Comments to the Author” section, enter your conflict of interest statement in the “Confidential to Editor” section, and submit your "Accept" recommendation.

Reviewer #2: (No Response)

Reviewer #3: All comments have been addressed

2. Is the manuscript technically sound, and do the data support the conclusions?

Reviewer #2: Yes

Reviewer #3: Yes

3. Has the statistical analysis been performed appropriately and rigorously? 

Reviewer #2: Yes

Reviewer #3: Yes

4. Have the authors made all data underlying the findings in their manuscript fully available?

Reviewer #2: Yes

Reviewer #3: Yes

5. Is the manuscript presented in an intelligible fashion and written in standard English?

Reviewer #2: Yes

Reviewer #3: Yes

6. Review Comments to the Author

Reviewer #3: Paper is now ready for publication. It's good to see that author have addressed all the suggested comments.

7. PLOS authors have the option to publish the peer review history of their article (what does this mean?). If published, this will include your full peer review and any attached files.

Reviewer #2: No

Reviewer #3: **Yes: **Dr. Gaurav Gupta
